# Bioerosion Research in the South China Sea: Scarce, Patchy and Unrepresentative

**Yen-Huei Li** [1] , **Barbara Calcinai** [2] , **Jiayi Lim** [1] **and Christine H. L. Schönberg** [1,3,]*

[1]  Department of Oceanography, National Sun Yat-sen University, Kaohsiung 80424, Taiwan
[2]  Department of Life and Environmenta Science, Polytechnic University of Marche, 60131 Ancona, Italy
[3]  School of Engineering, Oceans Institute, the University of Western Australia, Crawley, WA 6009, Australia
[*]  Correspondence: cschoenberg@mail.nsysu.edu.tw

**Abstract:** Coral reefs are in decline globally, resulting in changed constructive and destructive processes. The South China Sea is a marginal sea that is of high biological importance, but also subjected to extreme local and global pressures. Yet, the regional calcium carbonate dynamics are not well understood, especially bioerosion. A literature search for research on bioerosion and bioeroders in the South China Sea found only 31 publications on bioerosion-related research and 22 biodiversity checklists that contained bioeroders, thus generating a paltry bibliography. Bioerosion research in the South China Sea is still undeveloped and reached only two publications per year over the last few years. Hong Kong is the hotspot of activities as measured in output and diversity of methods, but the research in Hong Kong and elsewhere was strongly favoring field surveys of sea urchins over other bioeroders. Overall, macroborers received almost equal attention as grazer-eroders, but interest in microborers was low. Almost 90% of the research was conducted by local workers, but 90% of the publications were still disseminated in English. Field surveys and laboratory analyses made up over 40% of the research, but experimental work was mostly missing and represents the largest, most important gap. A government initiative in Thailand generated much knowledge on the distribution of marine sponges; otherwise urchins were again prominent in diversity checklists. Comparatively, many checklists were produced for Vietnam from work by visiting scientists. Most studies investigated coastal habitats, but a fourth sampled at oceanic locations. About 36% of the checklist publications covered the entire South China Sea; the rest produced faunistic records for locations within single countries. Our efforts demonstrate that, while active bioerosion research and basic expertise exist in the South China Sea, research remained unrepresentative with respect to taxa, ecofunctional guilds, and especially to controlled experiments. The latter are urgently needed for prognoses, modelling and management in this populated and overused marine environment.

**Keywords:** Western Pacific; literature review; bibliography; research focus; coral reef; disturbance

## 1. Introduction

Global climate change affects the world's environments at a rate that is thought to overwhelm many species and biotic communities before they can adapt to it, e.g., [1,2]. In this context, coral reef health is an increasing concern, as many nations are immediately dependent on this habitat, e.g., in 2014 the livelihood of 6 million people tied into coral reef fisheries, e.g., [3,4]. At the same time, there are increased reports of bleaching and mortality events, diseases, community shifts and unsustainable exploitation of marine habitats, e.g., [5]. If we want to slow this development and reverse it, we need to know all of the factors that affect the health of marine habitats, including coral reefs. The coral reef dynamic equilibrium is a proxy for reef health and represents a balance between accretion and erosion, e.g., [6–8]. Erosional processes can be significantly aggravated by global climate change, such as after heat-related coral mortality, e.g., [9]. Previous studies usually focused on calcification rates and ecophysiological functions of calcifiers, e.g., [10,11], but

in order to understand the present trends, we also have to assess the status of erosion. Erosion acts on reefs chemically as calcium carbonate dissolution, and physically in the form of breakage and relocation, either in the form of coral dislodgement or fragmentation or as sediment transport, e.g., [9]. Another component of reef erosion is biologically driven erosion [9,12,13]. The bioeroder community is made up of endolithic microborers (e.g., algae, bacteria, fungi) and macroborers (e.g., worms, sponges, molluscs) that create holes within the substrate and weaken it, whereas grazer-eroders (e.g., molluscs, urchins, fish) act on the substrate surface and wear the material down, e.g., [13]. Bioerosion is a process that has a larger effect on warm-water reefs than chemical dissolution, and, unlike physical erosion, it is permanent and continuous [9]. It is thus of central importance in reef health and structuring.

In the marine environment, a large focus for bioerosion lies on the carbonate materials of tropical coral reefs in the Pacific Ocean, and many bioerosion studies became available from this ocean [14]. Within the Pacific Ocean, there are areas where bioerosion is well studied (such as the Great Barrier Reef and the Mexican Pacific, e.g., [15,16], but other areas remain virtually unstudied. The South China Sea appears to be such a neglected area [17]. This is unfortunate, because the South China Sea is not just important in the context of its natural environment, it is also heavily used by anthropogenic activity. It is a marginal sea surrounded by ten densely populated and rapidly developing countries: Brunei, Cambodia, China, Indonesia, Malaysia, the Philippines, Singapore, Taiwan, Thailand, and Vietnam (Figure 1), e.g., [17]. These stakeholder countries strongly rely on marine environments and coral reefs for their livelihoods and food, and have reduced the system's resilience through overuse [18,19]. Damaging activities in the South China Sea include overfishing and overcollecting, destructive fishing, intensive aquaculture, coral mining, oil and gas extraction, land reclamation, coastal construction, pollution with debris and chemicals, eutrophication, military activities, intensive shipping traffic, and tourism, e.g., [20–29]. These local but serious impacts are increasingly overlaid with the effects of global climate change, leading to storm damage, reduced coral growth, mortality events due to heat stress, and the spreading of nuisance species and diseases, among other consequences, e.g., [30–34]. Moreover, several marine areas and islands are claimed by different nations, which increases the race for resources and strategic footholds, while it decreases opportunity and access for research, management and protection [22,35]. As a result, coral reefs in the South China Sea have suffered dramatic decline and experienced significant loss of live coral cover, driving many local reefs into an erosional state [36–39]. This situation is unfortunate, due to the large ecological value of this region. The South China Sea is located on the western margin of the Coral Triangle (Figure 1) and is inhabited by a remarkable diversity of marine life, including over 570 coral and over 3360 fish species [40,41]. In summary, the South China Sea is simultaneously a very important bioregion and a heavily exploited and largely disturbed ecosystem.

Due to the importance of this marginal sea and the diverse interests of its nations, we need to understand bioerosion processes in the South China Sea. For example, elevated levels of bioerosion can aggravate environmental changes that affect habitats that protect the coastlines and harbor commercial species, e.g., [8]. Also, detecting changes in the bioeroder community or in the severity of bioerosion can provide a range of information about the nature of environmental change, e.g., [9]. In order to detect changes in bioerosion, the organisms have to be correctly identified, and their basic ecophysiology should be understood. As a start, we need to know what information is presently available for bioerosion in the South China Sea and where the biggest gaps are. For this purpose, we conducted a detailed literature review to evaluate the current status of local bioerosion research and to identify knowledge gaps. We also performed a quantitative and qualitative analysis of the research as portrayed by the literature we retrieved, which we also list as a bibliography (Supplementary Materials).

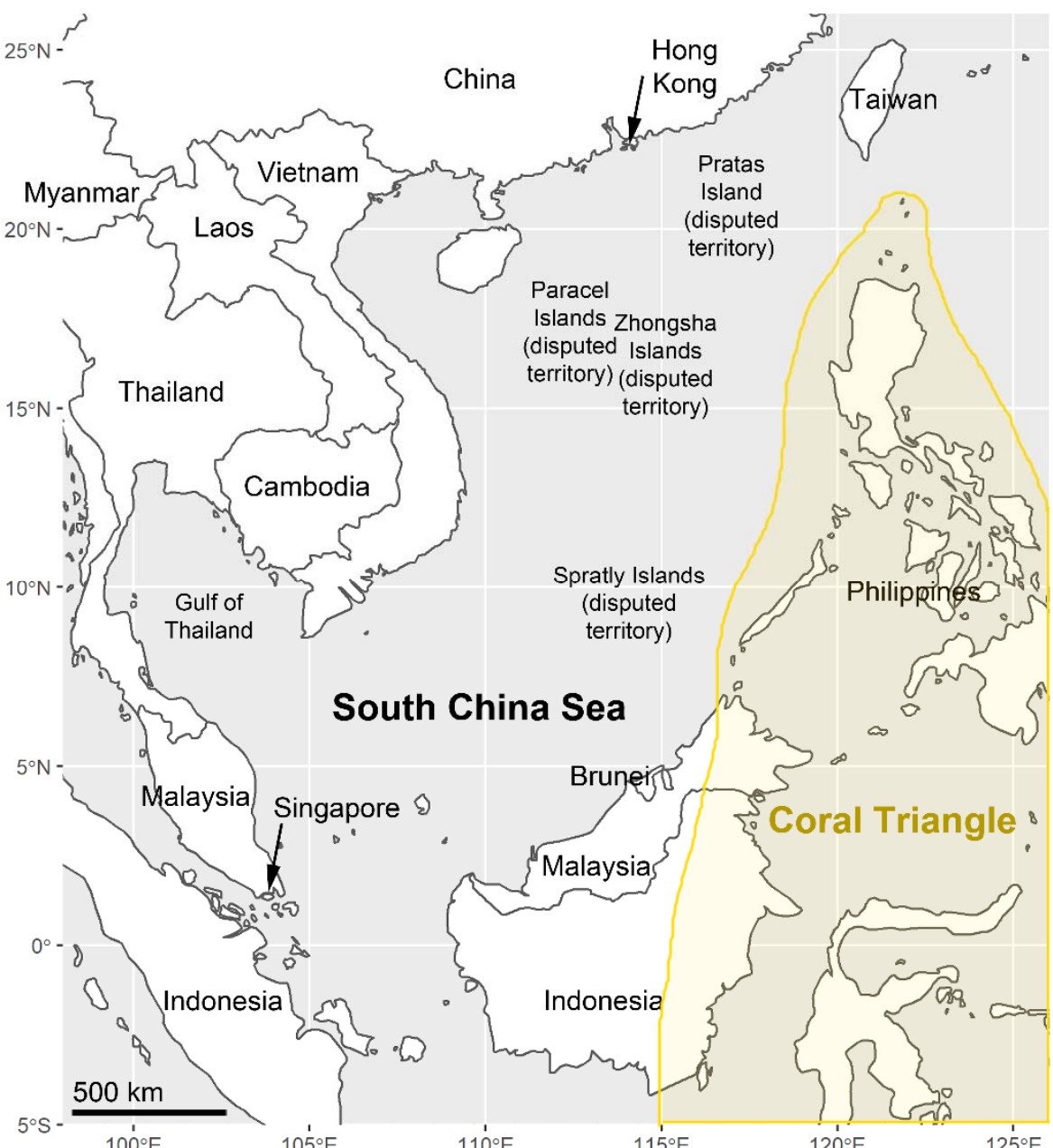

**Figure 1.** Map of the South China Sea, surrounded by the coasts of the stakeholder countries. Major disputed territories are marked in oceanic areas of the South China Sea. The area neighbors the Coral Triangle (highlighted in yellow).

## 2. Materials and Methods

Our study covers the South China Sea as shown in Figure 1 and includes the Gulf of Thailand. To assess the quantity and quality of bioerosion research in the South China Sea, we conducted a literature search in Google Scholar, excluding patents and citations [42]. Google Scholar finds mostly peer-reviewed, but also grey, literature. We screened the hits and omitted paleontological records, concentrating on recent biota. We used the keyword string "Bioerosion AND "South China Sea"". While some of the authors are native speakers of Mandarin Chinese, the search was conducted only with English keywords. Despite this, non-English publications were returned via matches with English titles, keywords, and abstracts included in non-English publications. Among such publications, papers in Mandarin Chinese were scored by members of the authorship team, and publications in

other Asian languages were scored using Google Translate [43]. In an attempt to assess how many non-English papers we may have missed, we translated "bioerosion" and "South China Sea" into Thai, Vietnamese and Chinese in Google Translate and repeated the search. In Thai, the first ten most relevant publications referred to water quality and terrestrial soil erosion, not to marine bioerosion; in Vietnamese and Chinese, to biodiversity studies. This suggested to us that either there were no more relevant publications, or that the technical term "bioerosion" may be challenging to translate and to search in other Asian languages, and that searches in local languages would incur significant additional effort that we regarded as beyond the scope of our publication. As we aimed to portray what literature is available to the "general user" who would conduct the search in English, we regarded our methods as suitable and the retrieved literature as representative.

The search yielded 820 hits that were screened until the content of the retrieved abstracts became irrelevant, e.g., when the keyword "bioerosion" appeared only in the reference list or was not part of the publication's main interest, or the publication was not from the South China Sea, etc. This happened after 200 references, and we stopped there. Reference lists in the publications we collected that way were also loosely screened, and promising titles that had not yet been found in the Google Scholar Search were also considered, but did not generate further material with a strict focus on bioerosion, e.g., the main topic was not on the process of bioerosion. Apart from our search in Google Scholar, we also searched the World Register of Marine Species (WoRMS, [44]). In our experience, Google Scholar is not an ideal search engine for taxonomic and historical publications. However, such literature is systematically deposited into WoRMS by the database editors as reference for taxon experts, and otherwise unreferenced work becomes available that way. We looked through this library with the button "literature" on the WoRMS starting page, using the search term "South China Sea" as contained in the title of the publication, which resulted in 553 references. WoRMS searches do not allow entering more than one search term to narrow down searches. The results therefore encompassed all taxa, not just bioeroders, and we went through this list and manually picked out relevant publications. Selecting suitable papers from the WoRMS-listed titles proceeded purely according to the taxa that were discussed in these papers, i.e., taxa that are known as bioeroders, which resulted in another small collection of titles (<25 publications). These papers were not on the process of bioerosion, but were relevant for establishing a knowledge on local faunistic diversity of bioeroders and were collated in table format.

Publications were viewed for context and grouped by the following six scoring categories that we used for a data analysis: (1) published year (in 5-year steps due to scarcity of data), (2) researcher's background (research institute, published language), (3) sampling site (country), (4) study design (field surveys/experiments or laboratory analyses/aquarium experiments), (5) bioeroder taxon group (fungi, algae, sponges, worms, bivalves, snails and chitons, crustaceans, etc.), (6) bioeroder type (microborer, macroborer, grazer-eroder, producer of homing and attachment scars, and shell drills; the latter three were scarce and were bundled as "other"). If a paper included information on more than one scoring category, we allocated each topic a partial score according to the percentage of contribution. For example, if a paper mentioned micro- and macroborers, as well as grazers, each would be scored with 0.33 so that the entire paper still added up to a score of 1. For this work, we did not consider publications that only mentioned or described bioeroders and did not further investigate their contributions to bioerosion. However, basic taxon lists can also be important when planning research in a designated area. We thus also collated and tabulated publications that provided faunistic checklists as a starting point for new research projects. This and the above information were then processed in Microsoft Excel Version 2205 to obtain figures for proportional relationships of the different situations per publication.

## 3. Results

### 3.1. Publication Yield and Publication Culture

A total of only 31 relevant publications on recent (non-paleontologic) bioerosion research in the South China Sea were retrieved from all our search efforts, screening the first 200 of 820 retrieved hits in Google Scholar, and 553 from WoRMS. Bioeroders were historically mentioned by authors describing species collected on expeditions or as reports from journeys in the South China Sea area, e.g., [45–47]. Regional investigations with a research focus on bioerosion processes only emerged very recently, with the first two publications appearing in the early 1980s (Figure 2). No further relevant publications could be found until 1999. After that, the number of papers on South China Sea bioeroders steadily rose, but total numbers stayed very low (Figure 2). In 2016–2020, the last complete 5-year period listed by us, only two publications per year were on bioerosion. The last period 2021–2025 is still incomplete and did not yet show whether the increasing trend will continue.

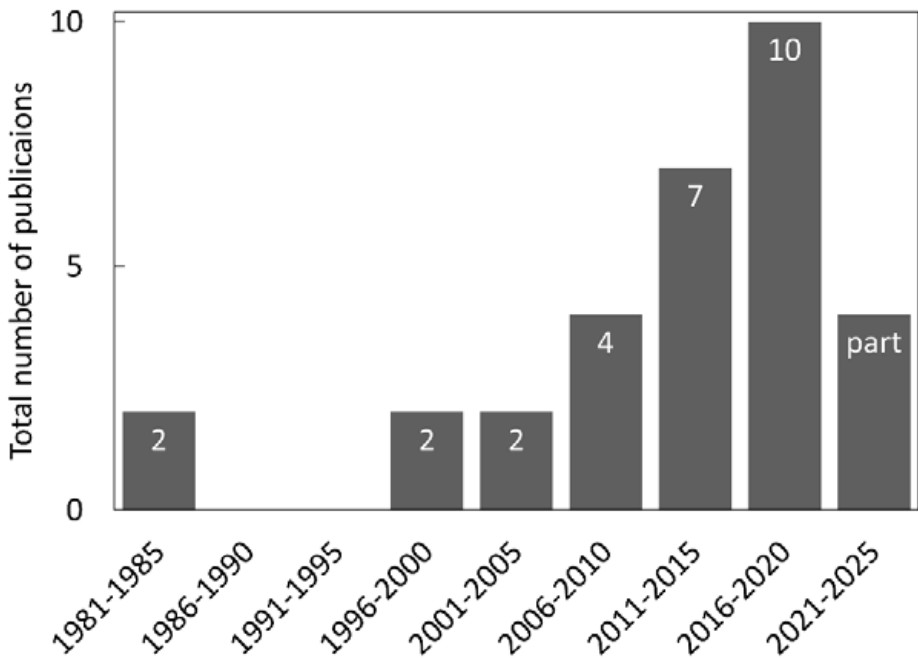

**Figure 2.** Number of publications on bioerosion research in the South China Sea over time in 5-year brackets, as based on a Google Scholar search using the search term '"bioerosion AND "South China Sea"" yielding 31 publications between 1982 and early 2022.

The retrieved studies originated in six different countries or were conducted in disputed territories (Figure 1). Almost half of them were from Hong Kong, which we are therefore showing separate location (Figure 3A; 45%). Other publications were from China (16%), Thailand (13%), disputed territory (13%, including the Spratly and Paracel Islands and places near the Zhongsha Islands), Vietnam (10%), and Malaysia (3%). Including Hong Kong in China, present China produced 61% of the publications. We found no publications on bioerosion processes from Taiwan, Singapore, the Philippines, Brunei, Cambodia, nor western Indonesia.

By far, most of this research (87%; Figure 3B) was published by local first authors. In contrast, 13% of the publications were contributed by foreign workers visiting the South China Sea during programs conducted from their own countries. Nevertheless, over 90% of the papers were written in English, 7% in Chinese, and the rest were in Thai (3%; Figure 3C).

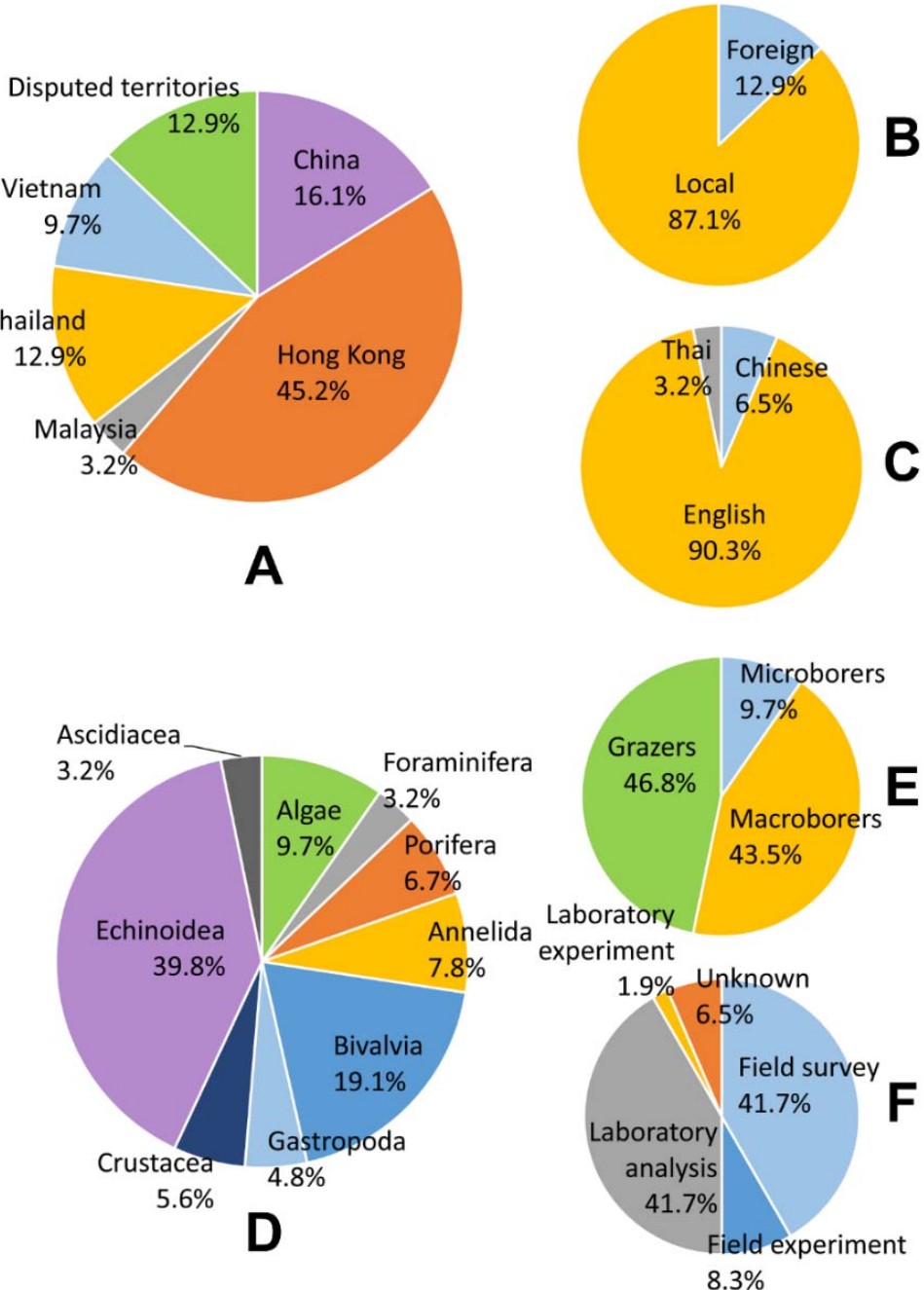

**Figure 3.** Proportional publications on bioerosion in the South China Sea based on a Google Scholar search using the search term 'bioerosion AND "South China Sea"', yielding 31 publications between 1982 and early 2022. (**A–C**) Proportional research culture. (**A**) Country of study or sampling. (**B**) Country of affiliation of first author. (**C**) Language of publication. (**D,E**) Proportional research target and method. (**D**) Studied taxon groups. (**E**) Bioeroder guilds, eco-function. (**F**) Research methods.

*3.2. Research Context*

Bioerosion research interest in the South China Sea was clearly dominated by investigations on sea urchins (Figure 3D). Overall, studies focused predominantly on epilithic grazers (47%; Figure 3E), which were mostly represented by sea urchins (40% urchins, 5% gastropods; Figure 3D). Endolithic macroborers received nearly as much attention as grazers (44%; Figure 3E), but the taxa in this bioeroder guild spread over more groups, with most interest invested into bivalves (19%; Figure 3D). Other common macroborers that were studied almost evenly divided into worms (mostly polychaetes, 8%), sponges (7%),

and crustaceans (barnacles, 6%). For practical reasons, we included two papers on unusual organisms in with the macroborers (3% each): one note on *Hyrrokkin sarcophaga* (unusually large foraminiferan attacking organisms from the surface, but penetrating quite deeply into the substrate), and one note on *Diplosoma* sp. (ascidian overgrowing and killing corals and apparently eroding the coral skeleton downwards). Microbial bioerosion data were published comparatively rarely, while grazer-eroders such as urchins and fishes received as much attention as endolithic macroborers: sponges, worms, and bivalves (Figure 3D,E). Within the macroborers, crustaceans and sponges were the least studied (Figure 3D). Apart from the urchin studies, hardly any publication identified bioeroders to the species level. Most of the authors used genus names or placed bioeroders into one of the categories we have used here.

Methods to study bioerosion in the South China Sea were divided into field surveys (45%; Figure 4C), laboratory analyses (38%), field experiments (8%), laboratory experiments (2%) and "unknown" approaches that could not be categorized by us (6%). This means that bioerosion experiments under controlled conditions are almost absent in the context of the South China Sea.

### 3.3. Data Cross-Comparison—Research Culture vs. Research Context

We recognized further patterns in the publications on bioerosion in the South China Sea. Cross-relating the different countries where sampling occurred with the respective research context revealed the leading role of Hong Kong in Southeast Asian bioerosion research. This was the only location where all categories of study design were performed, i.e., field and laboratory work, and observations and experiments (Figure 4A). In all other countries, a maximum of two study designs was pursued: field surveys and laboratory analyses.

China displayed the highest research diversity for taxon groups, with efforts almost evenly spread across microboring algae and the macroborers: sponges, bivalves, worms, arthropods and the ascidiacean *Diplosoma* sp. and *H. sarcophaga* (Figure 4B). Scientists from Hong Kong published on four macrobiotic borers and grazers, strongly dominated by research on sea urchins and bivalves, but also representing gastropods and worms. Urchins were also a research focus in Thailand and Vietnam, while disputed territories were the only study sites where algal microborers received much research interest. There was one Malaysian publication that evenly covered the three main macroborers: sponges, worms and bivalves.

When assessing the methods that were used to study the different taxon groups, urchins were the most comprehensively studied taxon, involving all study designs categorized by us (laboratory and fieldwork, observational and analytical approaches; Figure 4C). Other grazers (gastropods) were predominantly observed in the field. The macroborers sponges, bivalves, worms and arthropods, as well as microboring algae, were mainly investigated by field surveys and laboratory analyses, not through experimental work. Foraminiferan bioerosion traces were only evaluated in the laboratory. No data could be found on South China Sea bioeroding rates quantified under controlled conditions in aquaria. We found a total of seven publications or 22.6% with bioerosion rates, but again most of these were for only urchins (values displayed in the bibliography in the Supplementary Materials).

### 3.4. Faunistic Studies on South China Sea Bioeroders

Apart from the publications on bioerosion processes, we retrieved 22 papers that contained faunistic lists for the South China Sea that included bioeroders. As this is also useful information for planning research, we tabulated this information, listing the organism groups that the checklists covered (Table 1). For the material that we had accessed in this context, most publications were on Porifera (52%; Figure 5A). Other fauna groups that were studied divided into sea urchins (16%), annelid worms (9%), algae, chitons, gastropods, bivalves and fishes (5% each).

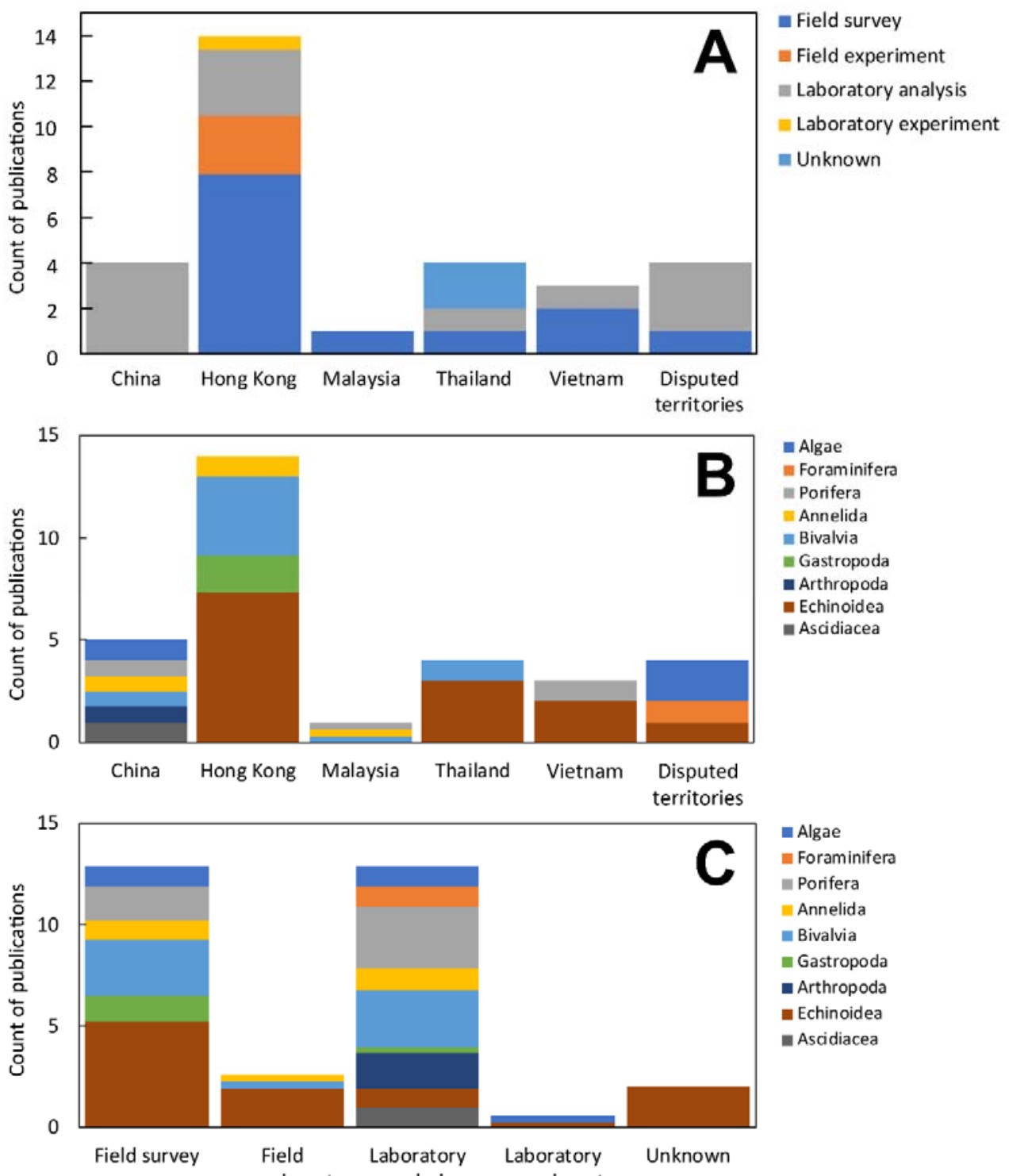

**Figure 4.** Cross-referenced data on proportional research on bioeroders in the South China Sea. (**A**) Approach of study in different countries. (**B**) Taxa studied in different countries. (**C**) Approach of study for different taxa.

**Table 1.** Faunistic checklists containing information on bioeroders in the South China Sea.

| References | Year | Taxon | Possible or Known Bioeroders | Study Area |
|---|---|---|---|---|
| [48] | 2007 | Porifera | Clionaidae, Spirastrellidae, Phloeodictyidae | Vietnam: Ha Long Bay |
| [49] | 2006 | Porifera | Clionaidae, Spirastrellidae, Phloeodictyidae | Vietnam: Ha Long Bay |
| [50] | 2021 | Porifera | Clionaidae, Phloeodictyidae | Vietnam: Ha Long Bay |
| [51] | 1984 | Algae | Hydrococcaceae, Entophysalidaceae, Anoplostomatidae, Haplosiphonaceae, Rivulariaceae, Gomontiaceae, Kommanniaceae, Phaeophilaceae, Ostreobiaceae, Delesseriaceae, Oscillatoriaceae (several nomina nuda were listed as well) | Spratly Islands (disputed) |
| [52] | 2006 | Echinoidea | Diadematidae, Toxopneustidae | Vietnam: Nha Trang Bay |
| [53] | 2016 | Annelida | Dorvilleidae, Eunicidae, Phascolosomatidae, Sabellidae, Serpulidae, Spionidae, Themistidae | South China Sea, Gulf of Thailand, Indonesia: Bangka and Belitung Islands |
| [54] | 2000 | Porifera | Spirastrellidae | South China Sea: Malaysia, Singapore, Thailand, Cambodia, Vietnam, Brunei, China (including Hong Kong), Philippines, disputed territories |
| [55] | 1998 | Echinoidea | Diadematidae, Echinometridae, Stomopneustidae, Toxopneustidae | Spratly Islands (disputed) |
| [56] | 2000 | Echinoidea | Cidaridae, Echinothuriidae, Diadematidae, Arbaciidae, Stomopneustidae, Temnopleuridae, Toxopneustidae, Parasaleniidae, Echinometridae, Strongylocentrotidae | South China Sea, Gulf of Thailand |
| [57] | 2016 | Porifera | Clionaidae, Spirastrellidae, Placospongiidae | Singapore, Malaysia, Thailand, Cambodia, Vietnam, Southern China, Taiwan |
| [58] | 2016 | Bivalva | Mytilidae, Pholadidae | Singapore, Malaysia, Gulf of Thailand, Vietnam, Southern China |
| [59] | 2000 | Annelida | Dorvilleidae, Eunicidae, Sabellidae, Serpulidae, Spionidae | China, Vietnam, Hong Kong, Taiwan, Singapore, Philippines, Thailand, Malaysia |
| [60] | 2007 | Porifera | Clionaidae, Spirastrellidae, Phloeodictyidae | Thailand: Had Khanom |
| [61] | 2011 | Porifera | Clionaidae, Spirastrellidae, Phloeodictyidae, Placospongiidae | Thailand: Chanthaburi and Trat Provinces |
| [62] | 2016 | Porifera | Clionaidae, Phloeodictyidae | Thailand: Mu Ko Tao |
| [63] | 2014 | Porifera / Echinoidea | Clionaidae, Phloeodictyidae / Diadematidae, Temnopleuridae, Toxopneustidae | Thailand: Mo Ko Samaesarn |
| [64] | 2013 | Porifera | Clionaidae, Spirastrellidae, Phloeodictyidae | Vietnam |
| [65] | 2000 | Pisces | Scaridae | South China Sea, Gulf of Thailand, Gulf of Tonkin |
| [66] | 2021 | Porifera | Clionaidae | Brunei: Pulau Bedukang |
| [67] | 2019 | Polyplacophora | Callochitonidae, Ischnochitonidae, Chitonidae, Mopaliidae, Acanthochitonidae, Cryptoplacidae | Guangxi, Guangdong, Hainan Island, Hong Kong, Xisha (Spratly) Islands, Dongsha Islands |
| [68] | 2001 | Gastropoda | Muricidae | Gulf of Thailand, Taiwan, Malaysia, China, Hong Kong |
| [69] | 2020 | Porifera | Clionaidae, Phloeodictyidae | Vietnam: Bai Tu Long, Ha Long Bay, Cat Ba and Ba Lua Archipelago |

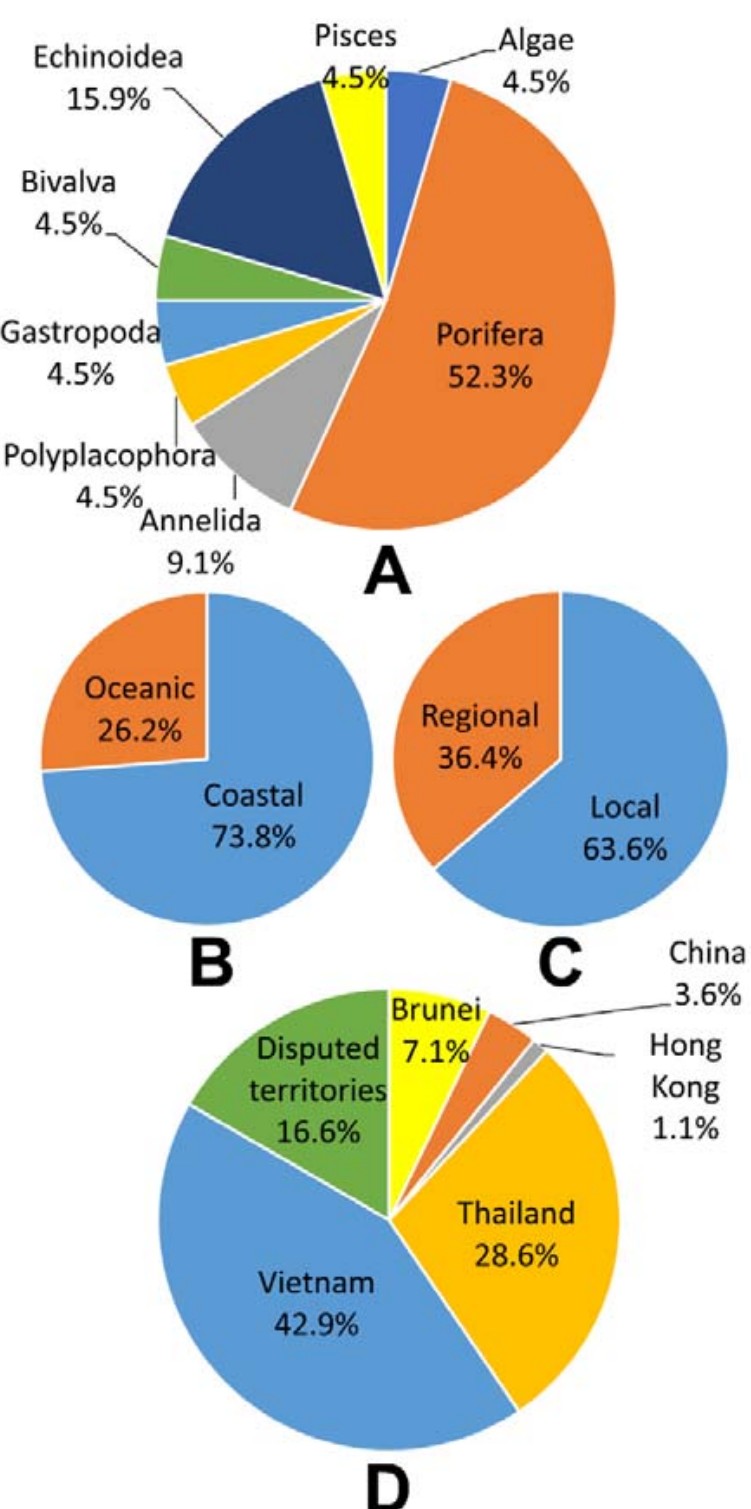

**Figure 5.** Proportional publications with faunistic checklists recording bioeroders in the South China Sea. (**A**) Taxon groups (n = 22). (**B**) Closeness of study location to mainland (n = 14; cutoff at 12 nautical miles). (**C**) Scale of research area, with regional (South China Sea) or local studies (within one country; n = 22). (**D**) Local faunistic checklists for sites within different countries (n = 14).

For the checklists, the research locations can be divided by distance to the mainland into coastal (74%) and oceanic studies (26%). We based the decision for the former within 12 nautical miles of territorial waters (1982 United Nations Convention on the Law of

the Sea; Figure 5B). Research further differed between "regional", i.e., concerning the whole South China Sea (36%) and "local" studies that remained confined to certain parts of countries (64%; Figure 5C). Within the local studies, most faunistic work was conducted in Vietnam (43%) and Thailand (29%; Figure 5D). Other publications were disputed territories (17%), Brunei (7%), China (4%) and Hong Kong (1%).

## 4. Discussion

Despite our best efforts, we found only 31 publications with a reasonably good relevance for bioerosion research in the South China Sea (bibliography attached as Supplementary Materials). There is also comparatively little published on local calcification [17], but any local or regional publications on coral reef health mostly referred to coral-related parameters such as coral cover, e.g., [21,70], or overfishing, e.g., [31], and increasingly also assessed microbial health, e.g., [71]. However, bioeroder-relevant factors were not usually investigated. This situation has previously been brought up in other publications [72], but this has not before been quantitatively assessed. Our literature review demonstrates that the lack of local bioerosion studies is significant and obvious. For comparison, we conducted our keyword search again for the Mediterranean, and within the first 200 hits we found twice as many publications for the Mediterranean than for the South China Sea, even though the Mediterranean Sea is smaller (2.5 milion km$^2$ for the Mediterranean, compared to 3.5 milion km$^2$ for the South China Sea; [73]). Moreover, the Mediterranean research had a stronger focus on bioerosion, i.e., published data were more process-oriented, and the Mediterranean studies involved more diverse approaches than what we had found for the South China Sea. In addition, bioeroder species were often identified in Mediterranean research, but not usually in South China Sea publications, where mostly genus names or bioeroder categories were used. The Mediterranean example search also confirmed for us that there was no problem with conducting the literature with search terms in English, because this search retrieved a large proportion of Italian, French and Spanish, as well as some Greek, Turkish and Russian publications. The outcomes clearly raise two large issues: the urgent need for bioerosion research in the South China Sea, and the need to provide quality species descriptions as basis for such work. At present, research is seriously hampered by the absence or patchiness of that knowledge. This is in striking contrast to the research need generated from the deterioration of coral reefs, a situation that will increase the incidence of bioerosion, e.g., [74].

Even within the few publications available, the research interest for various bioeroder taxa is strongly skewed, favoring sea urchins (Figure 3D). Urchins represent the best-known bioeroders in the South China Sea, but they are only one group in a diverse assemblage of epi- and endolithic bioeroders, e.g., [8,9]. We need to establish baseline knowledge and more comprehensive insights into the biology of other common and effective bioeroders, such as parrotfishes, sponges and bivalves, as well as microborers. However, this is where we encounter unknown invertebrate species and need reliable biodiversity checklists, as well as descriptions that display in situ characters of these organisms. Only then can we collect distribution data over time and larger scales, data that will help us understand bioerosion processes in the South China Sea. In part, related services are provided by Reef Check activities, which generate survey data over time, e.g., [75]. Yet, while corals and fishes are recorded at the species or genus level, bioeroders are again only reported at the coarsest levels or as "other benthos", if at all. This is a problem that exists in other marine environments as well [76]. This precludes monitoring of the negative side of the dynamic balance of coral reef construction and thus prevents early recognition of changes towards erosional states [11]. Of course, in some areas of the South China Sea, coral reef environmental monitoring has only recently been initiated. For example, in China, the first training was conducted in 2000 and organized by Reef Check Hong Kong [77].

Another issue in South China Sea bioerosion research is the blatant lack of controlled experiments (Figure 3F). Such experimental work can isolate response values per taxon group, as well as by ecophysiological environment and requirement, and can thereby quan-

tify rates under different conditions. This is an established method to predict bioerosion rates under climate change scenarios and into the future, e.g., [78–80]. Experimentally derived bioerosion rates are also used in comparison to calcification rates to assess local carbonate budgets and whether respective habitats are still positively calcifying or slipping into an erosional, deteriorating state, e.g., [81,82]. Such data are further vital in modelling coral reef health, e.g., [7,83–85]. The standard of existing local aquarium facilities can be limiting and may discourage local workers from attempting such experiments (authors' pers. obs.). Perhaps, for similar reasons, the approaches chosen for field surveys and laboratory analyses also remained simple, and included transect line surveys to count individuals and taxa, or basic dissection of samples.

It is therefore encouraging that researchers, especially Hong Kong researchers, pursue comparatively diverse methods and topics related to bioerosion, and prepare the area for others. In this, Hong Kong occupies a unique and leading position, and its research priorities in bioerosion differ quite markedly from those of China in gneral (Figure 4A,B). However, Hong Kong's marine habitats are supporting corals in environmental conditions that are naturally marginal for coral growth, e.g., [86,87]. It would be important to include more tropical reefs into bioerosion research in the South China Sea, so that conditions and developments can be mapped and projected into the future for the entire bioregion, and suitable management recommendations can be made.

It is thus also reassuring that the little research there is in the South China Sea is predominantly conducted by local researchers, who best know the local environment (Figure 3B). The research contribution by foreign workers is significant at 13%, but some local expertise is available, and the capacity is growing (Figure 2). The predominant publication language we found in the context was English (Figure 3C). We do not think that this a direct outcome of our search with only-English search terms, but we cannot guarantee that we may not have missed a small amount of non-English publications and may have found slightly different proportions if we had conducted additional searches in Chinese, Khmer, Thai, Vietnamese, etc. A small percentage of about 6% of the papers were published in Chinese, but some of these more or less duplicated the content of English publications of the same authors, e.g., [88–90]. In that way, this information was at least in part accessible to international users.

Our proportional breakdown of information available in faunistic checklists from the South China Sea was biased towards sponge research. This was largely caused by an initiative in Thailand that specifically supports biodiversity studies conducted by early-career scientists, and that thereby provided many active works on benthic communities and sponges, e.g., [60–63]. This is also the reason why Thailand had such a large proportional input into South China Sea bioeroder diversity research (Figure 5D). Ideally, such programs should also be implemented in other countries around the South China Sea, and in disputed areas. Faunistic research was strongest in Vietnam, however, and this was largely due to foreign programs for visiting scientists. In addition, while Hong Kong and China have a large influence on bioerosion research in the South China Sea in general, their efforts are targeting only few species and not faunistic surveys or collections. As the coasts of the South China Sea are rapidly changing, e.g., [91–93], we need to know more about available species diversities, and about which places need to be protected.

In summary, we clearly show that bioerosion studies are scarce, patchy and biased in the South China Sea region (Table 2). We acknowledge that the study of bioerosion is still a budding science for South China Sea researchers, and we need more quality species descriptions, numerous field surveys to assess bioeroder roles, and experimental studies to understand their responses to changing environments in Southeast Asia. Regarding the profound bias towards sea urchins, we encourage work on microborers, macroborers and parrotfishes as counterbalance. Collaborations would be of benefit, among local researchers, as well as between local and international experts. The "map" of South China Sea bioerosion research is still largely unexplored.

**Table 2.** Published research on bioerosion research in the South China Sea 1980 to present. Summary of the findings based on our literature analysis (see Supplementary Materials), highlighting bias, gaps and resulting research needs.

| South China Sea Bioerosion Research | Strengths | Weaknesses |
|---|---|---|
| Bioeroder taxa | Urchins (process) Sponges (diversity) | Fish, molluscs, worms, microbes |
| Bioeroder group | Macrobiota | Microbiota |
| Research sites | Hong Kong | e.g., Taiwan |
| Research methods | Observation | Experiments and hypothesis testing |
| | ⇓ | ⇓ |
| | Near-unlimited research opportunities | Overall lack in regional taxonomic knowledge at the species level, especially for invertebrates and microbes Lack of process knowledge at the ecophysiological level of local organisms, preventing predictions for locally dominant bioeroders Lack of large-scale research and time series, preventing trend recognition ⇓ Very limited opportunities for monitoring, recognition of change, for management, restoration and prevention |

**Supplementary Materials:** The following supporting information can be downloaded at: https://www.mdpi.com/article/10.3390/oceans4010005/s1, Table S1: Bioerosion research in the South China Sea—bibliography and scoring data.

**Author Contributions:** Conceptualization and methodology: C.H.L.S.; Literature search: all authors; Data analysis and figures: mainly Y.-H.L., supported by J.L.; Writing: all authors, with Y.-H.L. producing the first version; supervision and acquisition of funding: C.H.L.S. and B.C. All authors have read and agreed to the published version of the manuscript.

**Funding:** The manuscript is part of the Taiwanese project Schoenberg-MOST 110-2611-M-110-007 "Bioerosion in changing environments—enhancing data validity and scope of application" funded by the Ministry of Science and Technology, Taiwan.

**Institutional Review Board Statement:** Not applicable.

**Informed Consent Statement:** Not applicable.

**Data Availability Statement:** The data presented in this study are available on request from the corresponding author.

**Acknowledgments:** We gratefully acknowledge the following sources of funding in support of this research: Ministry of Science and Technology, Taiwan. We thank Benny K.K. Chan (Biodiversity Research Center, Academia Sinica, Taiwan) for providing literature. We further thank the reviewers, who improved our publication through their feedback.

**Conflicts of Interest:** The authors declare no conflict of interest. The project is government-funded, and the nature of the funding had no effect on the on the way data were retrieved, analyzed and discussed.

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
