# Peer review of "Bioerosion Research in the South China Sea: Scarce, Patchy and Unrepresentative"

_2673-1924, doi:10.3390/oceans4010005_

Round 1

Reviewer 1 Report

The title

The use of “scarce” and “patchy” seems to be repetitive. I would suggest to use either.

Lines 35, 66

Please specify what “global change” is referred to. 

The term “global change” has only been briefly mentioned in Introduction. I would suggest the authors to discuss on the potential “global change” impacts on bioerosion/calcification in the South China Sea.

Lines 62-64

“Damaging activities in the South China Sea include overfishing and overcollecting, destructive fishing, aquaculture, coral mining, oil and gas extraction, land reclamation, coastal construction, pollution and contamination, eutrophication, military activities, intensive shipping traffic, and tourism…”

“Overcollecting” is too general that the meaning has been covered by some of the other examples.

“Aquaculture” should not be considered as a damaging activity, unless it is too intensive and not sustainable.

As for “pollution and contamination”, please use either, since pollution is a category of contamination, but I would suggest to remove both, since they are not damaging activities, but the outcomes of these activities.

I have no further comment in the main text.

In Discussion or Conclusion, I would suggest the authors to provide a bullet-point summary for the future research directions.

Lines 350-351

“We thank to Benny K.K. Chan (Biodiversity Research Center, Academia Sinica, Taiwan) for literature supportion…” Is “supportion” a typo please?

Reviewer 2 Report

The paper is a review of bioerosion research in the South China Sea and highlights a need for this work in the region. The one point of confusion was how the different languages in the region were considered. Only English search terms were listed in the methods, yet language was one of the criteria assessed and returned three languages? Is this because some non-English publications were returned with English search terms (keywords, abstracts in English) or because other non-English search terms were used? Would the former be a representative sample of all the non-English publications in the region?

This feeds into larger issue of languages in global science. I suggest looking into and incorporating some of the work of Dr Tatsuya Amano and his translatE project: https://translatesciences.com/. Based on the paper in its current form, it seems only English search terms were used that also returned some non-English publications. If this is the case, I recommend clarifying the mechanism of this as best possible (inclusion of English keywords?) and clarify that the authorship team/who had the language abilities to read and assess the non-English publications included. Additionally, state that including search terms or search resources in the languages of all countries in the region was outside the scope of the study, but this also means that potentially relevant non-English works were likely missed. More comments can be found in the pdf.

Overall, this was a well conceived and written paper. It is a nice study that highlights gaps in bioerosion research in the South China Sea and a pressing need for faunistic studies as well.

Reviewer 3 Report

The presented manuscript aims to present a review regarding the research and knowledge of bioerosion in the southern region of China. However, I consider that in its current version, it represents more of a summary or state of the art rather than a review, which should consist of a critical analysis of existing published literature in a field through summary, analysis, and comparison, often identifying specific gaps or problems and providing recommendations for future research. A proper review also includes an analysis of the qualitative aspects of the manuscripts (site of collection, or type of approximation), and the methods. In addition, the manuscript covers in a very brief way the understanding of the bioerosion process or its relevance, what is the role of this process and its relevance, as well as who are the organisms that bioerode. The authors describe reef degradation (mostly causes), however, they do not focus on how knowing bioerosion (the process) will have a fundamental role in the conservation of their ecosystems. Therefore, I can not recommend the publication of this manuscript in the present form.

Specific comments:

Abstract:

Line 10. Include the relevance of bioerosion

Line 19-20. The authors talk about macrobioeroders, but they do not mention the type of bioeroders.

Line 26-28. Which are the most relevant contribution of this work, how does the research had contributed (or there is no contribution??) to the knowledge or need to do further research regarding bioerosion.

Introduction.

General comment. The introduction should focus on the bioerosion's relevance and process (this is briefly described in lines 41-46), rather than on the decline of the coral reefs. This section should include information regarding the type/classification of organisms involved (external and internal, as also macro/micro) and why there is an increase on the awareness of the effect of the bioerosion, which are the regional and local factors that may exceed the erosion and compromise the reef (both natural and anthropogenic)?.

Line 36. The organisms may adapt or cope/resist/ aclimatize. Please check if adapt is the proper term.

Line 54. Mexican Pacific should be Eastern Tropical Pacific

Materials and methods.

Line 96. How do the authors decide on the term “good focus”?

Line 103-106. This is not consistent. How the authors decide the relevant literature, and why do they discard references?

Results:

Line 141-143. The results are redundant with the figures. Given the low amount of data, the authors describe everything in the text, and it is repeated in the figure, so it is recommended to eliminate figure 3 except for 3d.

Line 153. The figure can be represented in a smaller table.

Line 159-163. If the research is focused on a few taxa, why do they not perform a deeper analysis of the results and their implications??

Line 172. Again redundant, the results section with the figure, and also is the first time they include the grazers, but they are not mentioned (characterized) in any section before.

Line 196-201. What is the scientific relevance of the studies? Is difficult to understand what is missing when an appropriate and critical analysis of the data available is not performed.

Line 212. The figure showed shallow information. Wich was the approximations for the field surveys or experiments?? What are the variables considered?? Each study includes only one organism, or does the research is focused on the interaction??

Line 225. The 12 nautical miles is base don?

Line 227. Local studies with contrasting conditions?

Line 230. The contribution of this information to the objective of the manuscript is not clear

Line 233. Again figure 5 has few data, which is repeated in the results main text.

Line 240-242. If the authors describe that there is data regarding calcification, which is important to understand the bioerosion, they did not show any data.

Line 250. What is better focus??

Line 256-257. How can they imply that there is not available dada when proper analysis is not performed and how can they imply that there will be an increase in bioersion??

Line 263. If the research has been focused in sea urchins, it should be important to describe the species and the results regarding that research, again, there is no data to support this idea.

Line 275. If there is a monitoring program and is important to mention it, which are the results?

Line 279-281. Using which stressors or scenarios??

Line 294. Why does this area is considered marginal?? Include quantitative and nod only qualitative data.

Line 304-306. Is not clear why this is relevant and how this will contribute to a new approaches into the study of bioerosion.

Reviewer 4 Report

General comments

The authors conducted an extensive literature search for research on bioerosion and bioeroders in the South China Sea to identify spatial, temporal and taxonomic patterns in bioerosion studies. The information is then used to identify spatial, taxonomic and methodological knowledge gaps for future bioerosion research to address. The study is generally well written, well presented and contributes important new knowledge on an understudies and highly threatened region. My major comment is that I would like to have seen actual bioerosion rates (or ranges) from previous studies included in the results and the discussion. Including bioerosion rates, at least for the for the major bioerosion groups, would help place bioerosion rates the south China sea in context with other regions and help identify areas of greatest uncertainly in bio-erosional processes in the south China sea, and thus future priority research areas.

Specific comments

Introduction

Line 47 – for consistency, consider defining biologically driven erosion, as done in the previous sentence for chemical and physical erosion.

Line 58 – Unclear what “ten populous” means? Suggest using alternative term or providing a population value

Sentence commencing line 60 needs supportive references

Line 66 – the word “overprinted” seems misplaced. Consider more commonly used English term such as “overlaid”

Line 78 – whilst this may be true, it needs to be made clearer why the “ importance of this marginal sea and the diverse interests of its nations” creates a need to understand bioerosion processes in the South China Sea? For example, how will an improved understanding of spatial, temporal and taxonomic patterns in bioerosion studies in the south China sea assist stakeholders, managers etc? Most likely in the identification of spatial and taxonomic knowledge gaps in bioerosion studies?

Materials and methods

Line 90 – 110 – The process for selecting articles needs to be made clearer so it can be reproduced.  For example what made an article become “irrelevant” line 92, What steps does “loosely screened” Line 94 require? What made an article seem “promising” Line 95. What terms or words were used to “manually pick out” Line 102 articles?

Line 104 – consider adding the list of known bioeroder taxa used in a supplementary file

Line 107 – Unclear what the “literature collection of the last author” refers to? Is this publications by this author? Consider clarifying the meaning

Line 11- what were the categories used to group literature?

Line 112 - what scoring category? There has been no mention of a scoring category in the methods to date? The scoring category should be explained first before discussing partial scores.

Results

Line 130 – Given only 31 articles these should be listed in a supplementary file

Lines 141 to 147 – Consider adding data discussed lines 141 to 147 to figure 1 as variable sized dots.

figure 1 – Excellent figure. A scale bar would be useful to provide better context for distances between stakeholder nations and reefs.

Figure 3a – consider showing this data as variable sized dots on figure 1 to provide greater understanding of spatial patterns in research.

Paragraph commencing line 175 - given the relatively few studies the authors should consider adding a list of the studies and specific detail on the methods and recorded bioerosion rates i.e. urchin densities / erosion range (0.5 - 2.5 kg.m2.yr). This information could easily be included in table form and would further help to identify methodological knowledge gaps for bioerosion research in the region.

Line 190 – grammatical error, pls check. “…where all from us counted….”

Line 192 – unclear what “research strategies” refers to. Consider revising

Discussion

Paragraph commencing line 238 - I would like to have seen actual bioerosion rates (or ranges) from previous studies included in the results and the discussion. Including bioerosion rates, at least for the for the major bioerosion groups, would help identify areas of greatest uncertainly in bioerosional processes in the south china sea and thus future priority research.

Figure 1 – excellent figures. Consider adding the spatial distribution data for studies  and a scale bar

Figure 3b – X-axis labels need editing so they do not overlap

Round 2

Reviewer 1 Report

All of my previous concerns have been addressed.

Author Response

Thanks for your suggestions and comments.

Reviewer 2 Report

This paper reviewing bioerosion research in the South China Sea has improved since the first round of revisions which is great to see. I do think it would would benefit from another round of editing - mostly writing. There are some very long sentences which are difficult to follow and should be broken up.

I thank the author for clarifying the methods section. It is much easier to follow now; however, some results (820 hits, etc) are included in the methods which should be in the results section. Thus, some reorganization is required.

I still take issue with some of the phrasing around English/non-English search returns. Mainly, the statement that because non-English publications were returned, the study is there for representative (of all research?). A truly representative search would include all the languages in the South China Sea. As noted in the paper, this is difficult and it is perfectly okay to say: we used English search terms and found some non-English publications with these search terms. Using additional non-English search terms was outside of the scope of the study although not doing so likely missed relevant research in the region. The lasted bolded part is necessary. There is a growing body of research (https://translatesciences.com/publications/) that shows the importance of non-English research publications in environmental sciences. The idea that all the English literature was searched, therefore our study is representative is a disservice to science especially when reviewing research in a region where English is generally not a national language of member countries. It takes a lot of conscious effort to combat this idea, but this paper provides a perfect opportunity to do so.

This work highlights important gaps in bioerosion research in the South China Sea. I look forward to seeing it published. Best of luck!

Detailed comments and suggestions are in the attached pdf.

Author Response

Thanks for your favorable review and your suggestion. We color-coded parts for easier recognition: blue – citations from our MS, green/italics – changes in our text, red – explanations where we rejected the proposed changes.

Reviewer 4 Report

This paper can be accepted.

Author Response

We appriciate your review

Round 3

Reviewer 2 Report

Please see attached review.
